# An agnostic study of associations between ABO and RhD blood group and phenome-wide disease risk

Torsten Dahlén[1,2]*, Mark Clements[3], Jingcheng Zhao[2], Martin L Olsson[4], Gustaf Edgren[2,5]

[1]Hematology Department, Karolinska University Hospital, Stockholm, Sweden; [2]Department of Medicine Solna, Clinical Epidemiology Division, Karolinska Institutet, Stockholm, Sweden; [3]Department of Medical Epidemiology and Biostatistics, Karolinska Institutet, Stockholm, Sweden; [4]Division of Hematology and Transfusion Medicine, Department of Laboratory Medicine, Lund University & Department of Clinical Immunology and Transfusion Medicine, Office of Medical Services, Region Skåne, Lund, Sweden; [5]Cardiology Department, Södersjukhuset, Stockholm, Sweden

## Abstract

**Background:** There are multiple known associations between the ABO and RhD blood groups and disease. No systematic population-based studies elucidating associations between a large number of disease categories and blood group have been conducted.
**Methods:** Using SCANDAT3-S, a comprehensive nationwide blood donation-transfusion database, we modeled outcomes for 1217 disease categories including 70 million person-years of follow-up, accruing from 5.1 million individuals.
**Results:** We discovered 49 and 1 associations between a disease and ABO and RhD blood groups, respectively, after adjustment for multiple testing. We identified new associations such as a decreased risk of kidney stones and blood group B as compared to blood group O. We also expanded previous knowledge on other associations such as pregnancy-induced hypertension and blood groups A and AB as compared to blood group O and RhD positive as compared to negative.
**Conclusions:** Our findings generate strong further support for previously known associations, but also indicate new interesting relations.
**Funding:** Swedish Research Council.

*For correspondence: torsten.dahlen@ki.se

Competing interests: The authors declare that no competing interests exist.

## Introduction

The blood group antigens of the ABO and RhD systems play a pivotal role in transfusion medicine because of their role in the safe administration of blood transfusions. In addition, these cell surface antigens have been demonstrated to have direct effects on the susceptibility for several diseases (*Franchini et al., 2012*; *Stowell and Stowell, 2019a*; *Stowell and Stowell, 2019b*). One of the first such studies was published in 1962, demonstrating a relationship between the ABO system and ischemic heart disease (*Bronte-Stewart et al., 1962*). Multiple subsequent studies have revealed associations with a range of diseases, with a prominent example being a decreased risk of thrombo-embolic events and increased risks of some hemorrhagic events in individuals with blood group O (*Franchini et al., 2012*; *Edgren et al., 2010*; *Vasan et al., 2016a*). The difference in thrombotic and hemorrhagic phenotypes has been attributed to variability in levels of *Factor VIII* and *von Willebrand factor*, where ABO status may explain as much as 30% of this variability (*Franchini et al., 2012*; *Orstavik et al., 1985*; *Germain et al., 2015*; *Lindström et al., 2019*; *O'Donnell et al., 2002*) Other

**eLife digest** The blood types that many people are familiar with, such as O-negative or AB-positive, are determined by two systems of antigens or proteins on the surface of the red blood cells: the ABO system and the RhD system. The ABO system types people's blood as A, B or AB if they have A and/or B antigens, or as type O if they have neither; while the RhD system provides the positive or negative label depending on whether or not the RhD antigen is present. Previous studies have found that some ABO blood groups are linked to increased risk and severity of a variety of conditions, including blood clots in veins, bleeding disorders and gastric ulcers.

Despite the known influence that blood groups can have on disease, the connection has not been fully studied in many conditions, particularly for RhD status. Knowing the differences in risk and disease severity between different populations could help clinicians identify individuals that they need to monitor more closely and include blood group information in prediction models.

To fill this gap in information, Dahlén et al. systematically looked for relationships between diseases and blood groups using records from 5.1 million people on a Swedish national blood donation-transfusion database. Examining 1,217 disease categories revealed that the vast majority did not appear to have a connection to either the ABO or RhD systems of classifying blood. However, the analysis identified 49 diseases with links to ABO blood types and one linked to RhD status. One notable finding was that people with blood group B have an decreased risk of kidney stones.

The distribution of blood groups varies significantly around the world, so this relationship between disease and blood group may in part explain regional differences in disease occurrence. In the future, identifying relationships with blood groups may help to better understand the underlying biological mechanisms of diseases and lead to new avenues of research.

prominent examples include associations with risks of a number of infectious diseases, to the extent that the allele distribution of the blood group antigens has evolved to reflect some areas endemic to these infectious diseases (*Franchini and Bonfanti, 2015*). This is in part true for infectious disease such as *Plasmodium falciparum* malaria, *Helicobacter pylori*, and *Vibrio cholera*, where ABO blood groups are involved in different aspects of pathogenesis, from microbe attachment and entry into cells to subsequent disease development and severity of disease (*Stowell and Stowell, 2019a*; *Franchini and Bonfanti, 2015*; *Cserti and Dzik, 2007*; *Degarege et al., 2019*). Surface antigens of the ABO blood groups are defined by the immunodominant, terminal sugar residues on certain glycolipids and glycoproteins anchored to the membrane of red blood cells and exposed extracellularly. Even if expressed from a single-gene locus, the *ABO* gene on chromosome 9, A and B antigens are present not only on erythroid cells but also in many other tissues, and due to the diverse tissue, expression may result in differences in disease occurrence (*Harmening, 2012*). The *A* and *B* allelic variants of the *ABO* locus encode the A and B glycosyltransferases, which differ only by a few amino acid residues, add the donor substrates UDP-*N*-acetylgalactosamine or UDP-galactose, respectively, to a common acceptor substrate, namely a carbohydrate chain terminating with the so-called H antigen, in turn dependent on fucosyltransferase activity expressed from the *FUT1* and *FUT2* genes on chromosome 19. In blood group O, the H antigen is left unaltered due to lack of ABO enzyme activity, most commonly by a single-nucleotide deletion in the *ABO* coding region (*Storry and Olsson, 2009*).

The RhD antigen, on the other hand, has a less clear link to health outcomes. RhD status has mainly been linked to alloimmunization of the pregnant women with hemolytic disease of the fetus and newborn as a consequence (*Urbaniak and Greiss, 2000*). Beyond these direct effects, little is known about its role in disease pathogenesis. The difference between RhD-positive and -negative blood group is the presence or absence of the RhD protein on the red blood cell surface. However, both individuals with and without RhD possess the homologous RhCE protein and Rh-associated glycoprotein (RhAG) on their red cells. Thus, functions carried out by RhD are likely performed RhCE and RhAG in RhD-negative individuals, and this redundancy may in part explain the scarcity of findings related to RhD status (*Avent and Reid, 2000*).

Using the Scandinavian Donation and Transfusion (SCANDAT) database, we have previously studied associations between ABO blood groups and cancer subtypes, cardiovascular and thromboembolic disease, the occurrence of dementia and degradation of bioprosthetic aortic valves in relation to ABO blood group (*Vasan et al., 2016a*; *Persson et al., 2019*; *Vasan et al., 2016b*; *Vasan et al., 2015*). However, these and most other prior studies into the association between ABO blood group and disease outcomes have been limited by potentially misdirected a priori hypotheses and phenome-wide disease associations have not been thoroughly explored in a systematic manner. Therefore, in the current study, we aimed to agnostically investigate the association between ABO and RhD blood group and disease occurrence for a large number of disease phenotypes using large-scale population-based Swedish healthcare registries.

## Materials and methods

### Study population and study design

Individuals in the study were identified using an updated version of the Scandinavian donations and transfusion database (SCANDAT3-S). This database includes close to 8 million individuals who have donated blood, received a blood transfusion, or have had blood group testing done for other reasons. Other reasons for blood group testing would typically be pre-emptive testing for example, before surgery or in antenatal care. The database contains detailed information on blood donations, transfusions, as well as blood group antigen and antibody testing results and is thoroughly described elsewhere (*Zhao et al., 2020*). It is nationally complete since 1995, but information dates back to 1968 with various levels of completeness, mainly depending on the geographical region. Using unique national registration numbers assigned to all inhabitants of Sweden, the SCANDAT3-S database has been linked to a range of national health outcomes registers, for hospital care, cancer, cause of death, and drug prescriptions . From SCANDAT3-S, we extracted information on ABO and RhD blood group and created a main cohort and a validation cohort. The main cohort consisted of all individuals who were born in Sweden where at least one parent was born in Sweden and who, for any reason, had undergone ABO and RhD blood group typing with a conclusive result, but who did not donate blood within 90 days of the test. Person-time for blood donors were excluded from the main cohort to maximize the representativeness of the study population. In the validation cohort, we included all individuals in the SCANDAT3-S database who had ever donated blood. As such, an individual could contribute person-time in both cohorts, such as in the case a person started to donate blood more than 90 days later from a blood grouping test that was initially performed for other reasons. The person-time before blood donation would contribute to the main cohort censoring at entry in the validation cohort starting at the time of blood grouping before the blood donation.

### Outcomes

We defined and studied a large number of disease categories. Non-cancer disease categories were based on discharge diagnoses from the national patient register, which covers all hospital inpatient care in Sweden since 1987 and all specialist outpatient care since 1997, and from the Cause of Death register, which records underlying causes of death for all persons in Sweden since 1964 (*Brooke et al., 2017*; *Ludvigsson et al., 2011*). Because the 10th revision of the International Classification of Disease (ICD) was implemented in 1997, we limited outcomes ascertainment to events from 1997 or later to avoid inconsistencies between ICD revisions. Cancer outcomes were based on the Cancer Register, which records all incident cancer cases in Sweden since 1958 (*Barlow et al., 2009*). All of these registries are held and maintained by the Swedish National Board of Health and Welfare and have a high level of completeness and accuracy. Dates of death and emigration were obtained from population registers kept by Statistics Sweden.

Details of non-cancerous disease categories are presented in *Supplementary file 3*. Non-cancer diseases were classified into disease categories based on the first three codes of the diagnosis, according to the ICD-10 codebook. We did not consider external causes of disease, traumatic injuries, or symptom-based codes as these were deemed unlikely to be related to blood group antigens. Cancer disease categories were based on anatomical coding using the 7th revision of the ICD for all non-hematological malignancies and the 8th revision of the ICD for hematological malignancies. For

details of cancer categories, see *Supplementary file 4* (SAS code for cancer disease grouping is available upon request).

In total, we considered 1217 distinct disease categories. After database construction, we excluded disease categories with fewer than 50 events before analysis as we would be unlikely to detect sufficient events in the validation cohort in categories with fewer than 50 events in the main cohort.

## Statistical methods

All persons were followed from the date of the first blood grouping test, from their 18th birthday, or from January 1, 1997, whichever occurred last. Follow-up was extended until the first incident event in each disease category, emigration, death or December 31, 2017, whichever occurred first. A person could thus be included in follow-up for all disease categories investigated.

Descriptive statistics were presented for cohort baseline data. For the main analysis, we used a Poisson regression model. In the model, we incorporated the following covariates: ABO blood group (A, AB, B or O), RhD status (weak or category expression variants were excluded), sex, calendar-period, and age. A restricted cubic spline functions with four or five knots placed according to Harrell's method were applied to the age and calendar-period covariates (*Harrell, 2015*). The regression model was fitted separately to each disease category resulting in incidence rate ratios (IRR) for each ABO blood group and RhD status using blood group O and RhD negative as reference, respectively. Wald's method was used to construct 95% confidence intervals (CI). Equi-dispersion was tested using a Lagrange multiplier test. For disease categories where data demonstrated significant over- or under-dispersion after also performing the same analysis but reducing the number of knots from 5 to 4, analyses were instead run using quasi-Poisson regression.

Multiple testing was handled using a two-stage approach. First, in the exploratory analysis using the main cohort, we applied a false discovery rate (FDR) adjustment of raw p-values assuming positive dependency of stochastic ordering between outcomes. Second, in the confirmatory analysis using the validation cohort, we used the disease categories with significant effects from explorative analysis, with results presented both without adjustment and using a Bonferroni adjustment. In effect, this allowed us to limit type one errors presenting confirmed associations with high certainty, but still not to compromise type two errors for future confirmatory analysis in other cohorts.

## Results

Characteristics of the main and validation cohorts are presented in *Table 1*. When combining the main and validation cohort, there were a total of 5.1 million unique individuals. The main cohort consisted of 4.2 million individuals who at any point had undertaken an ABO and RhD blood antigen test. The distribution of A, AB, B, and O were 47%, 5%, 10%, and 38%, respectively, and 84% of

**Table 1.** Baseline characteristics of main and validation cohort.

| | Main cohort | | Validation cohort | |
|---|---|---|---|---|
| Number | 4,204,234 | | 1,197,522 | |
| Age, median (IQR) | | | | |
| Age, median (IQR) | 52 | (30-71) | 30 | (23-41) |
| Year of birth, median (IQR) | 1949 | (1931–1971) | 1966 | (1953–1978) |
| Sex, % | 60 | | 49 | |
| Blood group, % | | | | |
| A | 47 | | 45 | |
| AB | 5 | | 5 | |
| B | 10 | | 11 | |
| O | 38 | | 39 | |
| RhD positive, % | 84 | | 82 | |

IQR, interquartile range.

individuals were RhD positive. Women constituted 60% of the cohort. The median age at cohort entry was 52 years (interquartile range [IQR], 30–71) and the median year of birth was 1949 (IQR, 1931–1971).

Not accounting for censoring due to disease events, the main cohort accrued a total of 49.9 million person-years of follow-up, 23.7 million in blood group A, 2.3 million in blood group AB, 4.9 million in blood group B, and 18.9 million in blood group O.

Of the original 1217 disease categories, 1090 remained available for analyses after excluding disease categories with fewer than 50 events. The median number of events per disease category in the main cohort was 4748 (IQR, 869–231,166). A meta-summary of results of regression analyses is presented in *Table 2*, and graphically in the form of a volcano plot in *Figure 1* (also, as an interactive, online variant as *Supplementary file 1*). Alternatively, results are also presented as an ICD chapter-based, variant Manhattan plot in *Figure 2* (also as an interactive, online variant as *Supplementary file 2*). Overall, in the main cohort and before FDR adjustment for multiple testing, there were 343 and 98 statistically significant associations for the ABO and RhD blood group systems and unique disease categories, respectively. Of these, a total of 143 (41%) and 13 (13%) associations between blood group and unique outcome remained statistically significant for ABO and RhD blood group systems, respectively, after FDR adjustment. For the ABO system, IRRs for statistically significant associations after FDR adjustment ranged from 0.57 to 0.99 for negative associations and from 1.01 to 1.52 for positive associations. For RhD status, IRRs ranged from 0.90 to 0.97 for negative associations and from 1.02 to 1.08 for positive associations. Details of all associations identified after FDR are presented in *Supplementary file 5* (for ABO blood groups) and *Supplementary file 6* (for RhD).

In our validation cohort, consisting of almost 1.2 million blood donors accruing 22 million person-years of follow-up, we validated the findings from the significant disease categories from the first analysis. Among the 143 and 5 significant disease categories for ABO and RhD, respectively, the median number of events was 7129 (IQR 2464–19,973). Before multiple testing adjustment, we identified 160 associations between a blood group in 143 and 5 disease categories, for the ABO and RhD blood group, respectively. After Bonferroni adjustment, there were 49 and 1 associations remaining between ABO and RhD blood group, respectively (*Table 2* and *Figure 3*).

**Table 2.** Meta-summary of results.

| | Main cohort | | | | | Validation cohort | | | | |
|---|---|---|---|---|---|---|---|---|---|---|
| Individuals | 4,204,234 | | | | | 1,197,522 | | | | |
| Person-years, sum | 50 M | | | | | 22 M | | | | |
| Events, median (IQR) | 4748 (869–23,166) | | | | | 7129 (2464–19,973) | | | | |
| | A | AB | B | ABO total | RhD positive | A | AB | B | ABO total | RhD positive |
| Before adjustment | 229 | 129 | 179 | 537 | 106 | 64 | 37 | 42 | 143 | 5 |
| Positive effects, N | 150 | 79 | 107 | 336 | 61 | 47 | 23 | 22 | 92 | 3 |
| Negative effects, N | 79 | 50 | 72 | 201 | 45 | 17 | 14 | 20 | 51 | 2 |
| After adjustment* | 108 | 56 | 70 | 234 | 13 | 26 | 13 | 10 | 49 | 1 |
| Positive effects, N | 66 | 38 | 36 | 140 | 10 | 19 | 10 | 6 | 35 | 1 |
| Negative effects, N | 42 | 18 | 34 | 94 | 3 | 7 | 3 | 4 | 14 | 0 |
| Positive IRR, median (range) | 1.05 (1.01–1.72) | 1.09 (1.03–1.52) | 1.09 (1.02–1.39) | | 1.05 (1.02–1.08) | 1.1 (1.03–1.57) | 1.32 (1.07–1.89) | 1.5 (1.07–1.64) | | 1.12 (1.12–1.12) |
| Negative IRR, median (range) | 0.95 (0.77–0.99) | 0.92 (0.74–0.97) | 0.92 (0.57–0.98) | | 0.97 (0.9–0.97) | 0.92 (0.86–0.95) | 0.84 (0.81–0.87) | 0.88 (0.83–0.93) | | - |

*In the main and validation cohort FDR and Bonferroni adjustment was conducted, respectively.

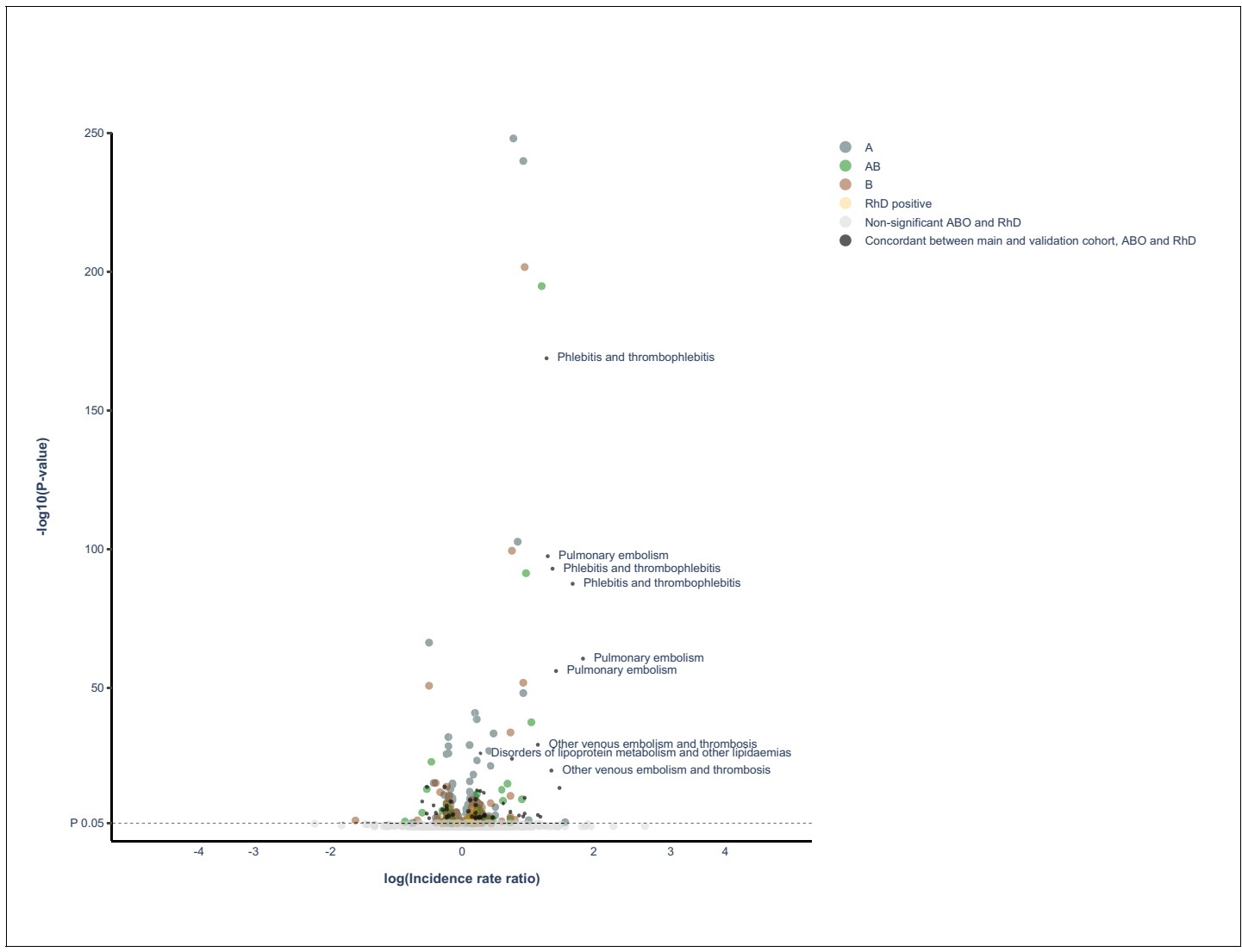

**Figure 1.** Volcano plot of all findings from main and validation cohort. Volcano plot depicting spread of p-values of significant and non-significant ABO and RhD blood groups for main cohort and validated results (live version available as online ***Supplementary file 1***). The labels represent the nine findings with the lowest p-value in the validation cohort.

A number of previously well-established associations were seen among the Bonferroni-adjusted results. For thrombosis, blood group A had a higher risk as compared to blood group O (e.g., pulmonary embolism, IRR 1.57 [95% CI, 1.51–1.64] and portal vein thrombosis, IRR 1.51 [95% CI, 1.25–1.83]). Bleeding disorders were more frequent in blood group O as compared to blood group A (e. g., gastric ulcer, IRR 0.92 [95% CI, 0.88–0.95] and duodenal ulcer, IRR 0.86 [95% CI, 0.82–0.9]). Thyrotoxicosis was also less common in blood groups A and AB as compared to blood group O (with IRRs of 0.90 [95% CI, 0.86–0.93] and 0.84 [95% CI, 0.77–0.92], for A and AB, respectively). Pregnancy-induced hypertension was less common in blood groups A and AB, as compared to blood group O (with IRRs of 0.95 [95% CI, 0.92–0.97] and 0.87 [95% CI, 0.83–0.92] for A and AB, respectively). Pancreatic cancer was the only malignancy that remained associated with a blood group, specifically blood group A as compared to blood group O (IRR, 1.29; 95% CI, 1.19–1.40). A new finding was that of calculus of the kidney and ureter, which were found to be less common in blood group B as compared to blood group O (IRR 0.93 [95% CI, 0.89–0.96]). Cholelithiasis, which has been disputed, was more common in blood groups A and AB as compared to blood group O (with IRRs of 1.07 [95% CI, 1.05–1.09] and 1.09 [95% CI, 1.05–1.13] for A and AB, respectively).

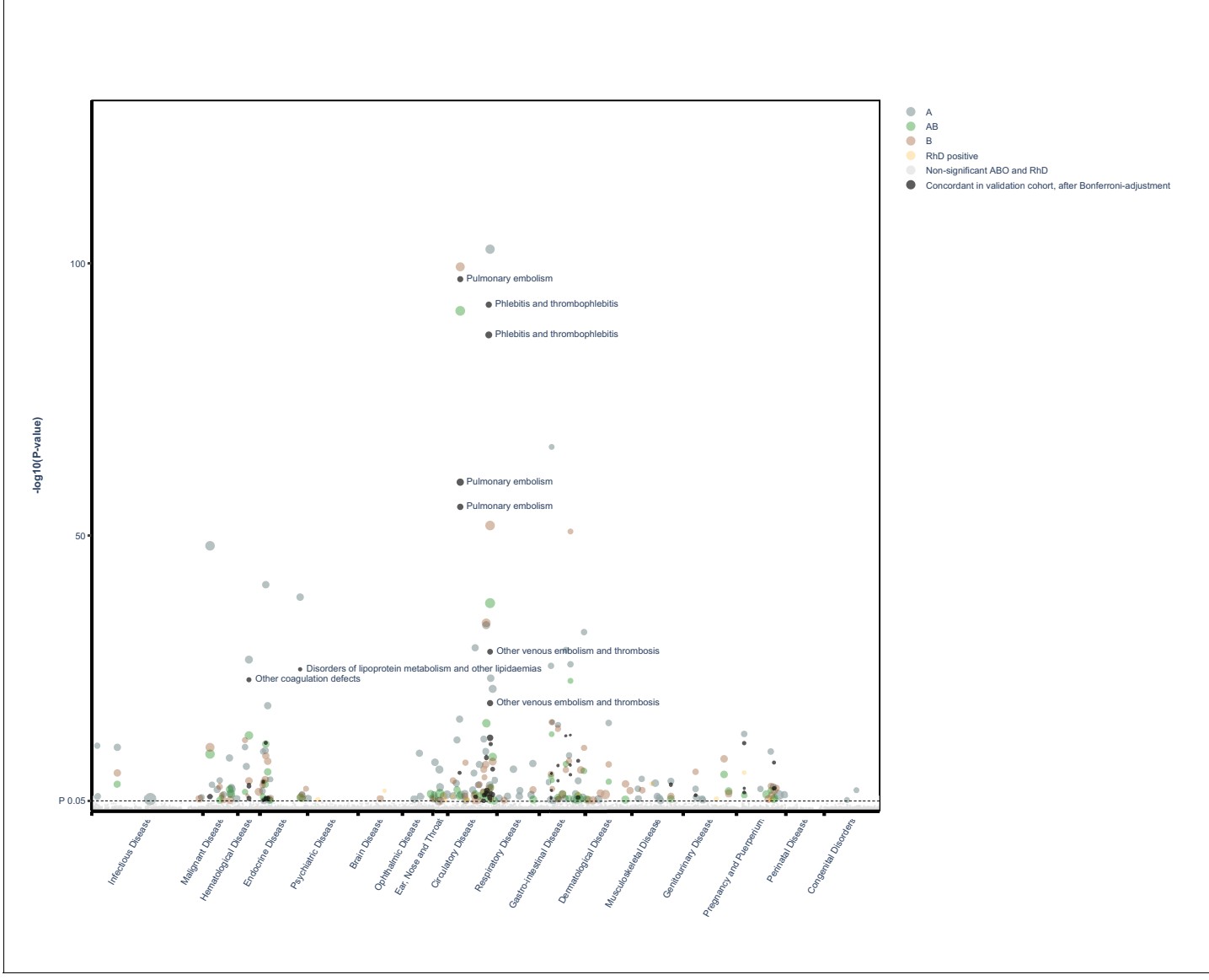

**Figure 2.** Manhattan plot of all findings from main and validation cohort mapped by ICD chapter. Manhattan plot depicting distribution of p-values for significant and non-significant associations between ABO and RhD blood groups and available outcomes for main cohort and validated results mapped by disease chapter in ICD (live version available as online *Supplementary file 2*).

In disease categories not significant after Bonferroni adjustment, some findings exhibited particularly strong effects, such as for viral and other specified intestinal infections, where blood group AB had a significantly lower risk, as compared to blood group O (IRR 0.74; 95% CI, 0.62–0.87). There was a lower risk for ankylosing spondylitis in blood group AB as compared to blood group O (IRR 0.79; 95% CI, 0.67–0.94), and for acute pancreatitis, again with a lower risk in blood group AB as compared to blood group O (IRR 1.14; 95% CI, 1.04–1.24). For the RhD positive as compared to RhD negative, only one disease category remained statistically significant after Bonferroni adjustment, namely pregnancy-induced hypertension (IRR 1.12; 95% CI, 1.09–1.16). Strong effects identified in the main cohort, but not in the validation cohort were hereditary factor VIII deficiency in blood group B (IRR 0.57; 95% CI, 0.42–0.77), well-differentiated thyroid cancer in blood groups AB (IRR 0.74; 95% CI, 0.62–0.88) and B (IRR 0.79; 95% CI, 0.70–0.90), measles in blood group A (IRR 1.72; 95% CI, 1.23–2.39), as well as both erythema nodosum (IRR 1.32; 95% CI, 1.15–1.53) and sarcoidosis in blood group B (IRR 1.15; 95% CI, 1.08–1.23), as compared to blood group O.

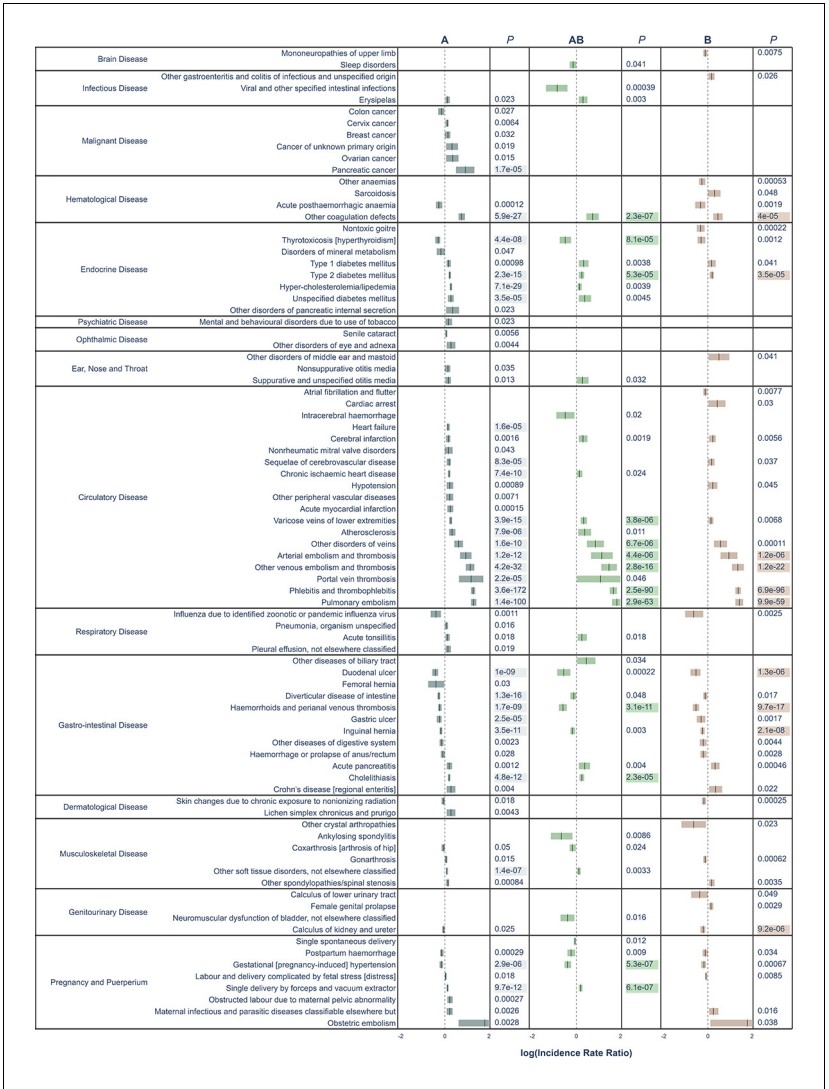

**Figure 3.** All significant un-adjusted findings from the validation cohort. Significant disease categories in the validation cohort. Blood group as compared to blood group O and log10(IRR) displayed with 95% confidence bands. All p-values are raw, highlighted p-value indicates associations that remained statistically significant also after Bonferroni adjustment.

## Discussion

In this large cohort study of 5.1 million unique persons followed over 70 million person-years, we performed an agnostic analysis of associations between the ABO and RhD blood groups and the risk of 1217 distinct disease categories. After multiple testing adjustment and comparison with a validation cohort, 49 and 1 associations between disease categories and blood group for ABO and RhD remained significant, respectively. Overall, we were able to confirm a number of previously known associations such as risk of thrombosis and hemorrhagic events. In addition, we also identified novel associations, some with firm evidence and valid even after conservative Bonferroni adjustment, in the validation cohort, including calculus of the kidney and ureter. Furthermore, being the largest study so far, we also found that blood groups A and AB had a lower risk of gestational hypertension, compared to blood group O, which has previously been disputed (*Clark and Wu, 2008*; *Franchini et al., 2016*; *Lee et al., 2012*). Of the identified associations, we speculate that some associations may be driven by increased screening due to other concomitant diseases that are associated with blood groups, which might be the case for hyperlipidemias that are screened for in heart disease. Most of the investigated disease or disease groups, however, do not seem to be strongly

influenced by the ABO blood group of the individual. As to possible mechanisms explaining the associations identified, we may only generate hypothesis to be further tested. It is however peculiar to identify a condition, such as renal caliculi, with highly variable distribution, in regard to geographic region, to have an association to blood group. A possible mechanism could be, when considering the full spectrum of disease categories investigated, that lower urine pH in diabetic patients, a disease category associated with an increased incidence in blood group B, may result in altered stone formation (*Sorokin et al., 2017*).

In previous studies, intending to dissect the association between blood groups and disease, there is diversity in findings and non-findings, something that may be a consequence of ethnic or geographic group, sample size, complexity of statistical modeling, or disease classifications used. This, however, imposes difficulties when comparing our results to the vast number of studies available concentrating the mentioned discoveries to a select few disease categories.

This is hitherto the largest study investigating blood group antigens and disease occurrence in an effort to find novel and confirm previously known associations. There are some particular strengths to our approach and data. Most notably, the study was based on a very large study population, representing one third of the Swedish population, with long-term and unbiased follow-up. This ensures both the reliability and the generalizability of the results. The agnostic approach also has the advantage of not being based on specific pre-set conceptions of specific disease categories and possible associations of blood group. All disease categories are treated equally and investigated using the same principles effectively removing researcher bias. Moreover, the data has been collected prospectively in various high-quality healthcare registries during a long time period with almost complete coverage. In addition, the fact that all blood group data in the SCANDAT3-S database were collected from clinical transfusion registers – the quality of which is essential for the safe administration of blood transfusions – ensures that there should be little or no errors in the blood group coding. Similarly, while the validity of the outcomes registration certainly varies between the different disease categories, the degree of such misclassification is unlikely to vary between blood groups, and so it should not affect the magnitude of point estimates.

The current study is limited by several factors. One such factor is the disease classification scheme used, based primarily on ICD revision 10 categories that in some instances may lack precision. Smaller disease entities were not accounted for and thus there may be true associations that were missed. It is thus possible that some of the associations that we reported were driven by multiple unknown associations within a specific disease category that may have unequal, or even detrimental, effects on the outcome. However, we believe that this limitation is an opportunity for further sub-categorized investigations in the future when even more events and follow-up time are available.

Another limitation that prevents strong casual inference is the possibility some of the observed associations between ABO blood group and disease categories were driven by other disease associations with ABO blood group. This might, for example, be the case for the associations between blood group A and diabetes as well as hyperlipidemia, which are potentially driven not by a causal association but possibly instead by an association between blood group A and ischemic heart disease, at the occurrence of which diabetes and lipid disorders are screened for and thus frequently diagnosed. We cannot exclude the possibility that some of the other associations were driven by similar non-causal mechanisms.

To limit the possibility of false-positive findings, we handled over-dispersed Poisson models using Quasi-Poisson and also in the main cohort applied the FDR approach, described by Benjamini and Hochberg, and then utilized a Bonferroni adjustment on the sub-grouped outcomes in the validation cohort. The aim of this approach was to reduce type one errors without being overly conservative by first conducting an explorative analysis in the main cohort. We also employed a confirmatory analysis with a validation cohort to further limit the possibility of false-positive findings. However, because the validation cohort was both smaller and consisted only of blood donors, who were selected for their good health, the ensuing smaller number of events may result in failure to detect potentially interesting associations. Furthermore, selecting explicitly healthy donors with no known serious health issue at time of start of donation may result in failure to detect less common or specifically non-acquired or early acquired disease. However, in the validation cohort, only approximately 1% of the categories had fewer events than 50 and all findings from the main cohort can be seen in the live *Supplementary files 1* and *2* and *Supplementary files 5–8*. It may still be informative to consider also some of the associations from the main cohort that were not corroborated in the validation

cohort. This is exemplified by pancreatic cancer where we saw an increased risk in blood groups AB and B in the main cohort (IRR 1.37 and 1.129, p<0.00001 and p<0.00001 for AB and B, respectively) and a similar, yet non-significant effect in the validation cohort (IRR 1.14 and 1.26, p-value 0.8 and 0.6 for B and AB, respectively). This also expands to the non-findings in terms of cancerous disease were multiple relationships that have previously been demonstrated, but not in the validation cohort after Bonferroni adjustment (*Vasan et al., 2016b*). This strengthens our decision to not limit the presentation of findings to only disease categories identified in the conservative Bonferroni-adjusted analysis.

Still, after these limitations we believe that our findings support and generate strong further evidence for previously known associations and indicate new and interesting relationships for disease such as calculus of the kidney and ureter, pregnancy-induced hypertension, well-differentiated thyroid cancer, and sarcoidosis. The new set of associations should be validated in other cohorts but also investigated using a mechanistic approach for a possible causal and biological interaction.

## Acknowledgements

The creation of the SCANDAT3-S database and the conduct of this study were made possible by a grant to Dr Edgren from Swedish Research Council (2017–01954). Dr Edgren is supported by Region Stockholm (clinical research appointment). Dr Dahlén is supported by Region Stockholm (clinical research appointment). Dr Zhao is supported by the Clinical Scientist Training Program and the Research Internship Program, both at Karolinska Institutet.

## Additional information

### Funding

| Funder | Grant reference number | Author |
|---|---|---|
| Swedish Research Council | 2017-01954 | Gustaf Edgren |
| Stockholms Läns Landsting | | Torsten Dahlén<br>Jingcheng Zhao<br>Gustaf Edgren |

The funders had no role in study design, data collection and interpretation, or the decision to submit the work for publication.

### Author contributions

Torsten Dahlén, Conceptualization, Data curation, Formal analysis, Funding acquisition, Investigation, Visualization, Methodology, Writing - original draft, Project administration, Writing - review and editing; Mark Clements, Martin L Olsson, Supervision, Methodology, Writing - review and editing; Jingcheng Zhao, Resources, Data curation, Methodology, Writing - review and editing; Gustaf Edgren, Resources, Formal analysis, Supervision, Funding acquisition, Validation, Investigation, Methodology, Writing - original draft, Writing - review and editing

### Author ORCIDs

Torsten Dahlén https://orcid.org/0000-0002-3856-7227
Jingcheng Zhao https://orcid.org/0000-0002-1776-1143
Martin L Olsson https://orcid.org/0000-0003-1647-9610
Gustaf Edgren https://orcid.org/0000-0002-2198-4745

### Ethics

Human subjects: The study has been approved by regional Stockholm County Board of Ethics Committee (ref nr: 2018/167-31). In Swedish register-based research informed consent, when involving a large number of individuals, does not need to be obtained.

Decision letter and Author response
Decision letter https://doi.org/10.7554/eLife.65658.sa1
Author response https://doi.org/10.7554/eLife.65658.sa2

# Additional files

## Supplementary files

• Supplementary file 1. Interactive Volcano plot depicting spread of p-values of significance and non-significance. ABO and RhD blood groups for main cohort and validated results. Labels are displayed on hoovering with information on blood group, disease category, p-value (FDR-adjusted and raw p-value depending on analysis, IRR with 95% confidence interval), events, and person-time.

• Supplementary file 2. Manhattan plot depicting distribution of p-values for significant and non-significant associations between ABO and RhD blood groups and available outcomes for main cohort and validated results mapped by disease chapter in ICD. Labels are displayed on hoovering with information on blood group, disease category, p-value (FDR-adjusted and raw p-value depending on analysis, IRR with 95% confidence interval), events, and person-time.

• Supplementary file 3. Non-cancer disease categories with ICD codes and names of categories, searchable html-file.

• Supplementary file 4. Cancer disease categories, searchable html-file.

• Supplementary file 5. All significant results from ABO analysis in the main cohort after FDR adjustment, searchable html-file.

• Supplementary file 6. All significant results from RhD analysis in the main cohort after FDR adjustment, searchable html-file.

• Supplementary file 7. All findings with p-value less than 0.05 in the validation cohort in the ABO analysis, Bonferroni robust findings labeled, searchable html-file.

• Supplementary file 8. All findings with p-value less than 0.05 in the validation cohort in the RhD analysis, Bonferroni robust findings labeled, searchable html-file and excel-file.

• Transparent reporting form

## Data availability

The patient level data used to construct the analyses cannot be made publicly available because of Swedish laws guarding the personal integrity of its citizens. Aggregate level data, which includes all the necessary information to recreate all the results can be requested from the authors, but requires IRB approval. This data includes subject blood group, age, sex, and calendar period, together with the corresponding number of person-years and the number of each type of event. IRB approval sought at the Swedish Ethical Review Authority (https://etikprovningsmyndigheten.se).

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
