## [Decision Letter]

**Acceptance summary:**

The connection between ABO/RhD blood type and the likelihood for a wide variety of human phenotypes is well established. This study is a comprehensive survey of 1,217 disease phenotypes in ~5 million individuals with known blood types. These data will provide a useful reference for many investigators.

**Decision letter after peer review:**

Thank you for submitting your article "An agnostic study of associations between ABO and RhD blood group and phenome-wide disease risk" for consideration by *eLife*. Your article has been reviewed by 2 peer reviewers, and the evaluation has been overseen by a David Ginsburg as the Reviewing Editor and Patricia Wittkopp as the Senior Editor. The following individual involved in review of your submission has agreed to reveal their identity: Karl C Desch (Reviewer #1).

Essential revisions:

1. How did you control for population structure? Are these associations driven by ABO directly or as a consequence of allele frequency differences in the population? The PheWAS equivalent of a QQ plot might be useful.

2. ABO blood group has been assessed using antigenic technics: This does not allow distinction of patients who are A1A1 vs A1O who can have different risks (at least this is what has been recently shown in venous thrombosis: PMID33512453).

3. A2 blood group is not so rare and might be associated with a risk than can be different from A1. It would be useful to add data specifically on A2 blood group.

4. The replication cohort is much smaller than the discovery cohort, which is typically suboptimal for an epidemiologic study. The authors should discuss this issue, and potential biases that could be introduced.

5. Although the discussion and results nicely point out the major findings, it might be useful to create a summary of novel findings and pertinent negatives (previous associations refuted) in a tabular form.

6. The paper would be strengthen by a literature search to elucidate potential mechanisms of association with renal calculi and other novel findings.

7. In the introduction it would be helpful to add 1 or 2 sentences explaining the ABO(H) locus and its relation to glycans on the cell surface.

8. The authors should note that the clinical phenotypes are derived from hospital coding and probably lack precision, especially in terms of diagnostic certainty.

---

## [Author Response]

Essential revisions:1. How did you control for population structure? Are these associations driven by ABO directly or as a consequence of allele frequency differences in the population? The PheWAS equivalent of a QQ plot might be useful.

Unlike in a GWAS study, with access to large numbers of genetic markers, we were unable to control for population structure using sophisticated statistical methods. Instead, as what we think is acceptable, we approached this by restricting the study population to individuals who were born in Sweden and who had at least one parent born in Sweden – which should limit the effects of population admixture given the relative homogeneity of the Swedish population (please see page 5, lines 93-95). We have, as per the suggestion of the reviewers, constructed a QQ plot to depict the relationship between observed and expected p-values for the exploratory cohort (Author response images 1 and 2).

**Author response image 1. sa2fig1:** ABO blood group. QQ plot for p-values in exploratory cohort in ABO analysis.

**Author response image 2. sa2fig2:** RhD blood group. QQ plot for p-values in exploratory cohort in RhD blood group.

2. ABO blood group has been assessed using antigenic technics: This does not allow distinction of patients who are A1A1 vs A1O who can have different risks (at least this is what has been recently shown in venous thrombosis: PMID33512453).

The reviewer is of course correct that, for example, A1A1 cannot be differentiated from A1O using serological (antigen- and antibody-determining) methods used in routine blood banking practice. That said, by combining family data (i.e. parents and offspring links which we have from the Swedish Multigeneration Register) that are available in the SCANDAT database together with blood group we are actually able to deduce more detailed blood group combinations (such as A1A1 and A1O), but only for a relatively small number of blood donors. Consequently, this interesting question was not statistically feasible to answer. In future updates of the SCANDAT database, with greater increases of population sizes, we hope to explore the A1 and A2 relationship more closely.

3. A2 blood group is not so rare and might be associated with a risk than can be different from A1. It would be useful to add data specifically on A2 blood group.

We agree that blood group A2 isn’t so rare, but A1-/A2-subtyping has only been done for a subset of our donors (<10%) and not for non-donors in our current database. As such, using our current methodology, we are unable to investigate the associations requested. We plan to conduct this in future updates of the database containing more information on A1 and A2 frequencies in donor cohort.

4. The replication cohort is much smaller than the discovery cohort, which is typically suboptimal for an epidemiologic study. The authors should discuss this issue, and potential biases that could be introduced.

Thank you for this excellent comment – we have expanded on the discussion as per your recommendations. We agree that, from a statistical point of view, it is suboptimal to have different population sizes of the two cohorts. Indeed, the choice of performing the analysis using this two-pronged approach was instead motivated by having two cohorts with completely different indications for ABO blood group testing, to avoid spurious results from such mechanisms. Also, we have decided to present all results, both un-adjusted results from the main cohort in the exploratory analysis together with the validation cohort, to be able to openly visualize possible associations that could be missed given difference in cohort size (please see supplementary live Figure 1 and 2). Another limitation, as discussed in the manuscript, is the fact that the validation cohort may only introduce disease that is acquired or discovered late in life due to the inclusion of donors without previously known health issues. This is now discussed on page 15, lines 281-288.

5. Although the discussion and results nicely point out the major findings, it might be useful to create a summary of novel findings and pertinent negatives (previous associations refuted) in a tabular form.

Very good suggestion. We have increased the discussion about potentially novel findings in the discussion (see page 13, lines 238-248). With regards to refuting previous associations, we are a little hesitant to go into too many details as it is inherently difficult with these types of phenome-wide studies to judge the extent to which we are truly powered to rule out associations. Therefore, we have not expanded the discussion about pertinent negative associations.

6. The paper would be strengthen by a literature search to elucidate potential mechanisms of association with renal calculi and other novel findings.

Thank you for the suggestion. This is now included in the manuscript, see page 13, lines 238-243.

7. In the introduction it would be helpful to add 1 or 2 sentences explaining the ABO(H) locus and its relation to glycans on the cell surface.

Thank you for the suggestion. We have added a section explaining the ABO(H) locus and its relation to glycans on the cell surface to introduce the reader to the broad mechanistic pathways that may alter disease occurrence depending on underlying blood group, see page 2, lines 51-61.

8. The authors should note that the clinical phenotypes are derived from hospital coding and probably lack precision, especially in terms of diagnostic certainty.

The clinical phenotypes are derived using hospital diagnoses from inpatient care and outpatient specialized care using the Swedish national patient register. The registry itself has been validated for a large set of outcomes referenced in the methods sections (Ludvigsson et al). However, as the reviewer points out, not all codes investigated have been validated in terms of their correlation to clinical disease and there is likely both under-, over- and misdiagnosis (i.e. misclassification). Still, the extent of such misclassification is unlikely to be related to patient ABO blood group status and will thus – if at all – bias results towards the null and/or only decrease power. As such, those associations that are detected are therefore unlikely to be driven solely by incorrect diagnoses. This has been addressed and clarified in the discussion (see page 14, lines 260-269).